# The Presence and Significance of Bacteria and Fungi in Bile Aspirated During ERC—A Retrospective Analysis

**DOI:** 10.3390/biomedicines13051255

**Published:** 2025-05-21

**Authors:** Sylvia Weigand, Arne Kandulski, Ina Zuber-Jerger, Marcus Scherer, Jens Werner, Jan Bornschein, Kilian Weigand

**Affiliations:** 1Department of Internal Medicine I, University Hospital Regensburg, 93053 Regensburg, Germany; 2Department of Surgery, University Hospital Regensburg, 93053 Regensburg, Germany; 3Experimental Medicine Division, University of Oxford, Oxford OX1 2JD, UK; 4Department of Gastroenterology, Gemeinschaftsklinikum Mittelrhein, 56073 Koblenz, Germany

**Keywords:** bile, bacterial colonization, bile aspiration, cholangitis, ERC, sphincterotomy

## Abstract

**Background:** Infections of the biliary tract are found frequently in pathologically or surgically altered bile ducts. Mostly these infections result from the ascent of bacteria or fungi from the small bowel, although hematogeneous infections of the biliary system may also occur. The biliary sphincter and the continuous flow of bile are thought to prevent or limit ascending infections. Obstructive alterations in the biliary system are the most frequent indication of endoscopic retrograde cholangiography (ERC). The aim of this study was to analyze the spectrum and frequency of microbes within the bile, and discover the influence of earlier sphincterotomy. **Methods:** In our department, we routinely aspirate bile for microbiologic culture during ERC. For this study, all ERC performed in 2014–2018 were retrospectively analyzed, including all microbiological reports. Indications for the endoscopic examination were also recorded. In addition, the findings were correlated with whether or not a sphincterotomy had been previously performed, and whether or not there had been antibiotic treatment prior to the examination. **Results:** A total of 2253 successful standard ERC procedures were performed between 2014 and 2016. In 486 cases, bile was aspirated and sent for microbiologic culture. In total, 1220 bile samples were analyzed, and bacteria or fungi were found in 1029 (86.0%). *Enterococci* and *Enterobacter* were found most commonly, but *E. coli*, *streptococci*, *klebsiella*, and *staphylococci* were also found. In 11.2% of positive cultures, multiresistant pathogens were identified. In up to 29% *Candida* spp., most commonly, *Candida albicans* (68%) were also found, either alone or in conjunction with bacteria. Neither prior sphincterotomy nor the use of peri-interventional antibiotics had a major influence on the frequency with which positive bile cultures were detected. **Conclusions:** Aspiration of bile during ERC is of high clinical relevance, because microbiological analysis reveals the frequent presence of bacteria and fungi, knowledge of which may be useful for deciding on anti-infective treatment.

## 1. Introduction

Cholangitis is a potentially severe disease in which bile bacteria or fungi translocate into the blood [1,2,3]. There are established criteria for the diagnosis and assessment of severity of acute cholangitis [4,5]. There is also a risk of abscess formation in the liver as a result of ascending cholangitis [6]. Infection is common in an altered biliary tract. It is reported that bacteria are found in the bile in up to 70% of patients with gallstones [7]. The most common reasons are obstruction due to bile stones or cancer, post-surgical changes, or chronically inflamed bile ducts [1,8,9]. Elevated intrabiliary pressure and reduction in anti-adherence factors seem to be the main causes of bacterial translocation [8]. Colonization and infection of the bile normally result from ascending infection by gut flora. It has been reported that an incompetent or resected sphincter of Oddi increases the risk of cholangitis [10,11]. A rare cause is hematogenous spread following infection of the biliary tract [11]. Aspiration of bile during an endoscopic retrograde cholangiopancreatography (ERCP) is the best way to examine the colonization and infection of the bile [1,12,13], including culture of the pathogen and resistance testing [14].

It is at present unclear if microorganisms are just colonizing the bile and at what time point, or for what reason this transforms into an infection requiring antibiotic/antifungal treatment. It is known that bacteremia frequently results from cholangitis, and that the spectrum of bacteria and fungi in the bile is wide and may change over time [2,14]. In addition, it has been shown that the bacterial species identified between 2010 and 2012 differ from those identified in earlier years and that resistance to certain antibiotics increased from 2007 to 2012 [2,9,15,16]. It is therefore very important to evaluate pathogens and resistance regularly, in order to ensure the use of the most appropriate antimicrobial therapy, should this become necessary [17]. This is especially important, given that there is no consensus regarding the best peri-interventional and/or initial antibiotic regimen in cholangitis [18,19,20].

A recent study has shown that the cultures from bile aspirated during ERCP in patients with suspected acute cholangitis were positive in 91,8% with increasing rates of multiresistant organisms [21]. Several possibilities for this high incidence of microbes in bile cultures are under research. Moreover, the risk of transmission from the oral cavity [22] or via duodenoscopy [23] most likely is ascending colonization via the biliary sphincter.

In the present study, we analyzed bacterial and fungal bile colonization over a 5-year period from 2013 to 2018 and the impact of antibiotic treatment and altered barrier function after sphincterotomy. Because we are a tertiary center, dealing with a high volume of chronic inflammatory bile diseases, liver oncology, hepatic surgery, and liver transplantation, we see a broad spectrum of patients undergoing ERCP for a variety of reasons. The aim of this study was to analyze the spectrum and frequency of microbes within the bile, and what is the influence of earlier sphincterotomy.

## 2. Material and Methods

### 2.1. Patients

This retrospective study was performed at the Department of Internal Medicine I at the University Hospital Regensburg. The study was approved on 4 July 2018 by the local ethics committee [No. 18-1051-104] and was performed according to the updated guidelines of good clinical practice and updated Declaration of Helsinki. All interventions were routine procedures and patients provided written informed consent prior to the procedure.

All patients undergoing examination of the bile system by endoscopic retrograde cholangiography (ERCP) from January 2014 to December 2018 were included in this study. We noted if bile had been aspirated for microbiological diagnostic, whether or not a sphincterotomy had been carried out previously, and if antibiotic treatment was provided peri-interventionally. Data were retrieved from the patient record and the clinical data information system. If bile had been aspirated, we noted if and what kind of microorganism was detected and if there was any major antibiotic resistance. The data were pseudonymized.

### 2.2. Endoscopy

ERCP was performed with Olympus duodenoscopes (TJF-160VR). Patients were routinely anesthetized for the procedure with propofol and midazolam. They were monitored by pulse oximeter and non-invasive blood pressure measurement during the endoscopy and for at least 30 min after the end of the treatment. All examinations were performed under CO_2_ insufflation. Bile was aspirated through an ERCP catheter introduced into the main bile duct. Aspiration was performed as a primary step after the intubation of the main bile tract before a sphincterotomy was performed or contrast media were administered.

### 2.3. Microbiological Assessment

The collected bile sample was sent unmodified in a vial to the microbiological department immediately following the ERC. There, it was cultured on different growth media, and bacterial identification was performed according to the standard laboratory protocol, comparable to the published protocols [22,24].

### 2.4. Statistical Analysis

Our study was retrospective. The results are therefore explorative and should be used to generate hypotheses. Statistical comparison of groups was performed by T test, where appropriate. Statistical significance was set at *p* ≤ 0.05. To compare the distribution of two cohorts, the Kolmogorov–Smirnov test was used. Statistical significance was set at *p* ≤ 0.05. To calculate the probability of identifying a bacterial species or group of bacteria between EPT+ and EPT− patients, the chi square test was used; statistical significance was set at *p* ≤ 0.05.

## 3. Results

### 3.1. Study Population and Characteristics

We included 2253 successful, standard, endoscopic retrograde cholangiographies (ERCs) from January 2014 to December 2018 in this study. All ERCs that were terminated were excluded because patients had not fasted, or because of respiratory distress before reaching the papilla or because of other patient conditions. Also excluded were ERCs performed by single-balloon enteroscopy or by a percutaneous approach, or because of other post-surgical changes in normal bile drainage. Mean age of the study population was 56.7 years with a range of 3 to 96 years. A total of 1463 (64.9%) of the patients examined were male. Further demographics and indications for ERC are shown in Table 1. All ERCs were performed by one of three experienced endoscopists. The frequency of bile aspiration during the ERCP was 45.3%, 77.4%, and 89.6%. Reasons not to aspirate bile during the examination were technical problems in bile aspiration, scheduled follow-up examination after a few days with bile aspirated in the first examination, and in most cases the decision of the endoscopist.

In total, 1220 bile samples were aspirated during ERCs for microbiological analysis. Altogether, 1197 (98.1%) of the bile fluids were analyzed in the microbiological department of our hospital, indicating a widespread interest among clinicians in the hospital, in having bile samples analyzed. Positive cultures were found in 1029 (86.0%) of bile aspirates.

### 3.2. Influence of Sphincterotomy

Because the main cause of cholangitis is ascending infection of the bile duct from the bowel, we investigated if earlier sphincterotomy had an influence on how often microbiological bile samples have become positive. To do so, we divided the cohort into two groups: patients with a sphincterotomy or precut performed at an earlier examination (EPT+) and patients with naive papilla (EPT−). Figure 1 shows the percentage of positive bile cultures over the five years analyzed. There were no significant differences between the five years. Interestingly, there was a strong increasing trend to aspirate and analyze bile over the five years. Microbiologic analysis of bile fluid increased progressively over the 5 years of the study (32.4%, 40.3%, 47.2%, 60.3% to 72.5%). There is an apparent trend toward more positive bile cultures in the EPT+ group in the years 2015, 2017, and 2018 (Figure 1). However, the difference of analyzed bile fluids between the two groups was >5% more in EPT+ than EPT− patients in these three years, whereas the difference was <3% in the other two years. Thus, there appears to be no significant difference in positive bile cultures between patients with or without prior sphincterotomy.

Comparing the number of different microorganisms in bile between EPT+ and EPT− patients (Figure 2), there is a trend toward fewer microorganisms in the bile of EPT− patients. However, this trend was not statistically significant for the total cohort (Figure 2a) or for individual years (Figure 2b–f).

### 3.3. Microbiological Analysis

Regarding the microbiological analysis, we first looked for changes in the distribution of the bacteria that were found over the period under analysis. Figure 3 shows the percentage of the different microorganisms found. For this analysis, we clustered microorganisms in different categories according to how often individual microorganisms were found. Furthermore, we distinguished between Gram-positive (greenish) and Gram-negative (blueish) bacteria. There was no change in either Gram-positive or Gram-negative bacteria over the five-year period. The only significant increase over the study period was in multiresistant bacteria. No atypical bacteria were detected in the bile of any of our patients.

Comparing the bacteria found in EPT+ with those found in EPT− patients, there were significant differences in the probability of identifying a certain bacteria/group of bacteria between the two groups (Figure 4). *Pseudomonas aeruginosa*, *Citrobacter freundii*, *Klebsiella oxytoca*, and *Escherichia coli* were significantly (*p* < 0.05) more often found in EPT+ patients. *Klebsiella variicola*, *Klebsiella pneumonia*, other *enterobacteria*, *Enterococcus faecalis*, *Enterococcus faecium*, and other *Enterococci* were very significantly more common in EPT+ patients (*p* < 0.001). No significant differences between groups were found for *Diplococci*, *Gram-positive rod bacteria*, *coccoid rods*, *other negative rod bacteria*, and other *Staphylococci*. Interestingly, in the EPT− group, *Staphylococcus epidermidis* was found significantly (*p* < 0.05) and *Streptococci* very significantly (*p* < 0.001) more often than in the EPT+ group. The distribution of the bacteria did not differ significantly between the individual years of the study period.

Multiresistant bacteria were highly significantly (*p* < 0.001) more common in the bile of EPT+ patients than in EPT− patients. The most common multiresistant microorganism was vancomycin-resistant *enterococcus* (VRE) in both EPT+ and EPT− patients (Figure 5). However, while in EPT+ patients multiresistant *Escherichia coli* was the second most common microorganism, no multiresistant *E. coli* was found in the bile of EPT− patients. In addition to VRE, only multiresistant *pseudomonas aeruginosa* was found in EPT− patients. In both cohorts, a clear increase in the various multiresistant bacteria was seen over the five-year study period (Figure 5).

### 3.4. Fungal Analysis

Fungal analysis yielded positive results in 28–29% of bile cultures from both cohorts, EPT− and EPT+. All fungi found were *Candida*. In both groups, *Candida albicans* was the most common, found in roughly 68% of fungi-positive cultures. The second most common fungus, *Candida glabrata*, was found in ca. 20% of samples from both cohorts (Figure 6). There was no difference between EPT+ and EPT− groups with respect to the detection of *Candida* in the bile.

### 3.5. Influence of Antibiotic Treatment

To assess the influence of antibiotic treatment, we assigned the bile aspiration population to one of four groups. First, we divided the population according to whether (52%) or not (48%) antibiotic treatment was provided. In so doing, we did not take into account whether antibiotic treatment was provided only in the peri-interventional period or long term for cholangitis or other reasons. Patients taking vancomycin or rifaximin orally only were included in the non-treated group. Second, we differentiated these two groups according to whether or not a sphincterotomy had been previously performed. Patients undergoing sphincterotomy during the examination were assigned to the groups without sphincterotomy. A total of 74.5% of the patients in the antibiotic-treated (ab+) group had a sphincterotomy (EPT+), 25.5% did not (EPT−). In the group without antibiotic treatment (ab-), 77.6% had a sphincterotomy, 22.4% had not. There was no statistical difference between these two groups.

Positive bile cultures were found in 91.1% of the ab+EPT+ group, 69.6% in the ab+EPT− groups, 85.2% in the ab-EPT+ group, and 65.9% in the ab-EPT− group. No significant difference was found between the two EPT+ and EPT− groups. However, there was a difference between the EPT+ and EPT− groups in the ab+ versus the ab- groups. Taken together, the results show that patients with sphincterotomy more often yield positive bile cultures, but antibiotic treatment does not result in a difference in the detection of microorganisms in bile.

To rule out the possibility that this was due to the differences in antibiotic treatment among the four subgroups, we compared the antibiotics used (Figure 7). The main antibiotics used were Piperacillin/Tazobactam, Meropenem, and Glycopeptides (mainly Vancomycin). There was no significant difference associated with the antibiotics used between the EPT+ and EPT− groups, even when comparing patients with positive or negative microbiological cultures (MiBi pos/neg).

## 4. Discussion

Cholangitis is one of the reasons for performing an ERC on patients, either to rectify the cause of cholangitis or to collect bile for culture and optimize antibiotic treatment [7,12]. The major reason for the infection of the bile system is ascending infection from the gut [10]. Therefore, it is most likely that microorganisms from the small bowel are the cause of cholangitis. However, at present, there is a paucity of data on which bacteria and fungi cause cholangitis and how commonly. Furthermore, after ERC, patients frequently develop bacteremia or septic episodes due to the wash-in of microorganisms colonizing the bile system, especially after therapeutic ERC [1,13]. In both cases, it is likely, that a sphincterotomy increases the risk of colonization and infection of the bile system. Up to now, there have been few investigations of these issues [11,25].

One recently published study by Gromski et al. analyzed the microbiology of bile aspirates only patients with suspected acute cholangitis [21]. The authors describe a high rate of positive cultures in these patients. In this study, prior sphincterotomy was reported and was associated with an increased frequency of some species and the probability of vancomycin-resistant *Enterococcus*. These data support our findings with the strength of our study that all patients were included, and so the colonization and infection of the bile duct are evaluated. Furthermore, recently published data investigate if the transmission of oral microbiota during the ERC or by possibly contaminated duodenoscopes may be the source of colonization [22,23]. However, there was no clear data supporting this hypothesis. The species found in these studies were also comparable to our spectrum and mainly represent gut flora.

There are other recent studies mainly investigating the resistance to antimicrobial therapy [21,25] or the increased 30-day mortality in patients with bacteriemic cholangitis [16]. Both studies showed positive bile cultures similar to those found in our study. The main aim of our study was to investigate the influence of sphincterotomy. In addition, we examined the influence of peri-interventional antibiotic treatment on the frequency of microorganism-positive bile cultures, colonized or infected. Furthermore, we also identified the microorganisms involved, in order to help with choosing the correct antibiotic treatment. This again fits with the recently published data [16,21,25].

There is considerable clinical interest in bile culture. Almost all aspirated bile fluids from all departments of our center were sent for cultural analysis. The decision to send bile for analysis was made by the attending doctor of the patient, and not by the endoscopist. Reflecting this interest, the numbers of bile samples collected increased greatly during the study period. This may also have been due to the fact that bacteria and fungi were detected in 86% of all bile fluid cultures. Thus, the colonization of bile is higher than previously reported in most studies. We were surprised to discover that the colonization of bile was comparably frequent in patients without prior sphincterotomy. There were fewer co-colonizations and slightly fewer positive cultures, but our findings indicate that the barrier function of the sphincter [11] is not as efficient as previously thought. Just a few of these patients had suspected cholangitis before presenting for ERC, but were examined due to other pathologies of the bile system.

Interestingly, antibiotic treatment also had a minor effect on the detection rate of the bacteria found in the bile cultures. This means that, despite antibiotic treatment, the aspiration of bile for microbiological assessment is a desirable additional procedure. Regarding the broad range of different bacteria we found in our patients, this information is useful to potentially adapt the antibiotic regimen in these patients. This is especially true for multiresistant bacteria, which showed significant increases in the bile cultures, even within the rather short study period of five years.

Not surprisingly, most of the bacteria found were enterococci and enterobacteria, but the full range of Gram-negative and Gram-positive bacteria was found in the bile of our patients. Thus, the ideal peri-interventional antibiotic treatment is hard to define but should be adapted in light of the findings from bile cultures. In the case of fungal analysis, only *Candida* spp. was found, mainly *Candida albicans*. Therefore, the treatment options are clearly defined. However, *Candida* spp. was found in a rather high proportion (28–29%) of our patients, and without the results of bile culture, we would not know which patients should be treated for *Candida* in the first place. Thus, bile culture can also help to identify patients in need of antimycotic treatment.

There are several limitations to this study. First of all, due to the retrospective character and the lack of clinical data, we cannot know if the microbes identified in the bile culture of our patients represent a colonization or a true infection of the bile system. Furthermore, the antibiotic treatments were not standardized with respect to the type of antibiotic or duration of treatment. Some patients were undergoing antibiotic treatment for a variety of infections and other patients were receiving peri-interventional prophylaxis. Thus, the data on the effects of antibiotics on microbiological cultures must be treated with caution. We did not differentiate between precut and sphincterotomy, and pooled them together into one group. However, there was no overall difference between these groups. Finally, we did not take into account of whether the bile culture was slightly positive (+) or strongly positive (+++) for certain microorganisms. The findings would therefore be affected by whether or not the patients received antibiotics. Furthermore, there may be an effect on the results depending on whether or not the patients had colonized or infected bile.

Our findings clearly show that aspirating bile during ERC is of clinical relevance. Very often, microbiological bile assessment reveals positive bacterial and fungal cultures. This may help to adapt anti-infective treatment in patients. Neither sphincterotomy nor peri-interventional antibiotic treatment has a major impact on whether or not microorganisms are detected or on which microorganisms are found. Thus, we suggest that the aspiration of bile should be performed routinely during ERC to provide microbiological profiles, should they be required to refine further treatment. Further studies maybe should only include specific underlying diseases and evaluate the influence of earlier sphincterotomy prospectively.

## Figures and Tables

**Figure 1 biomedicines-13-01255-f001:**
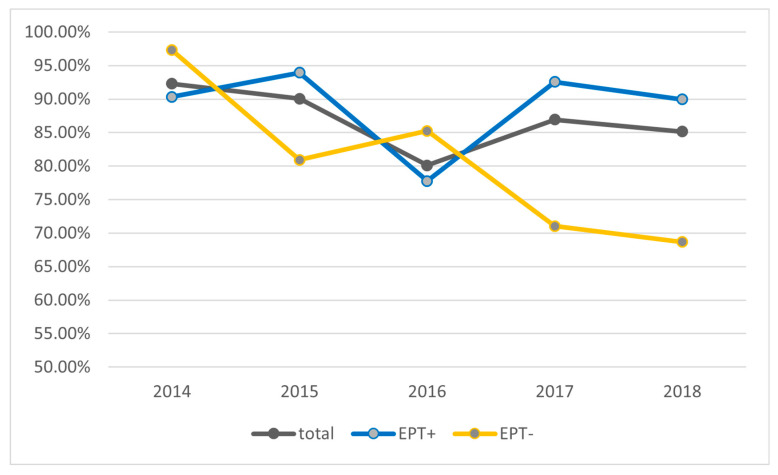
Percentage of bile cultures with positive microbiological findings. Shown are the total cohort as well as the groups with or without prior sphincterotomy (EPT) over the study period of five years.

**Figure 2 biomedicines-13-01255-f002:**
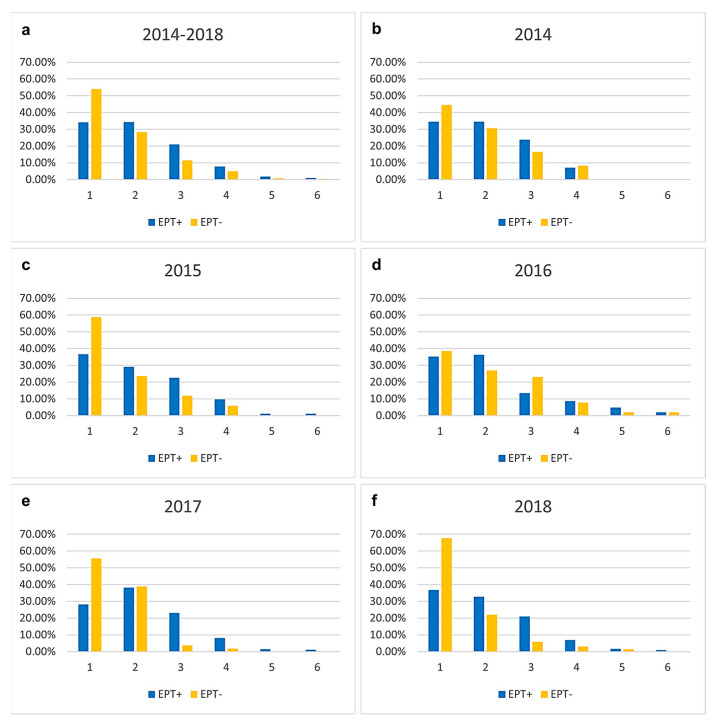
Number of different microorganisms found in bile in the total cohort (**a**) and separated for every year (**b**–**f**). Results are presented in percentage separated into the two groups of individuals with and without sphincterotomy (EPT) at an earlier examination. There is a trend toward fewer microorganisms in the bile of EPT− patients. However, this trend was not statistically significant.

**Figure 3 biomedicines-13-01255-f003:**
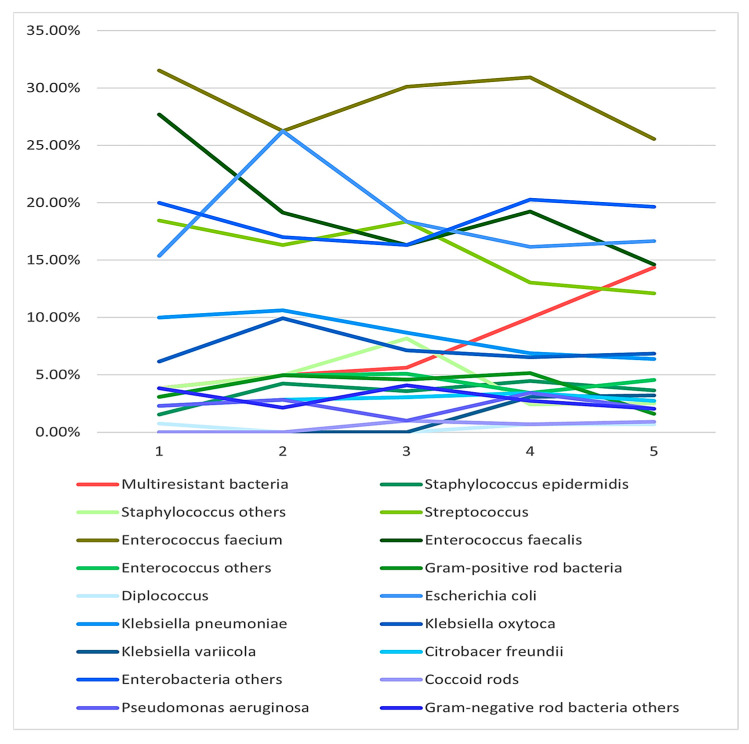
Different groups of bacteria found in bile cultures over the five-year study period. Multiresistant bacteria showed a significant increase over the five years, whereas there was no significant change in the number of other bacteria.

**Figure 4 biomedicines-13-01255-f004:**
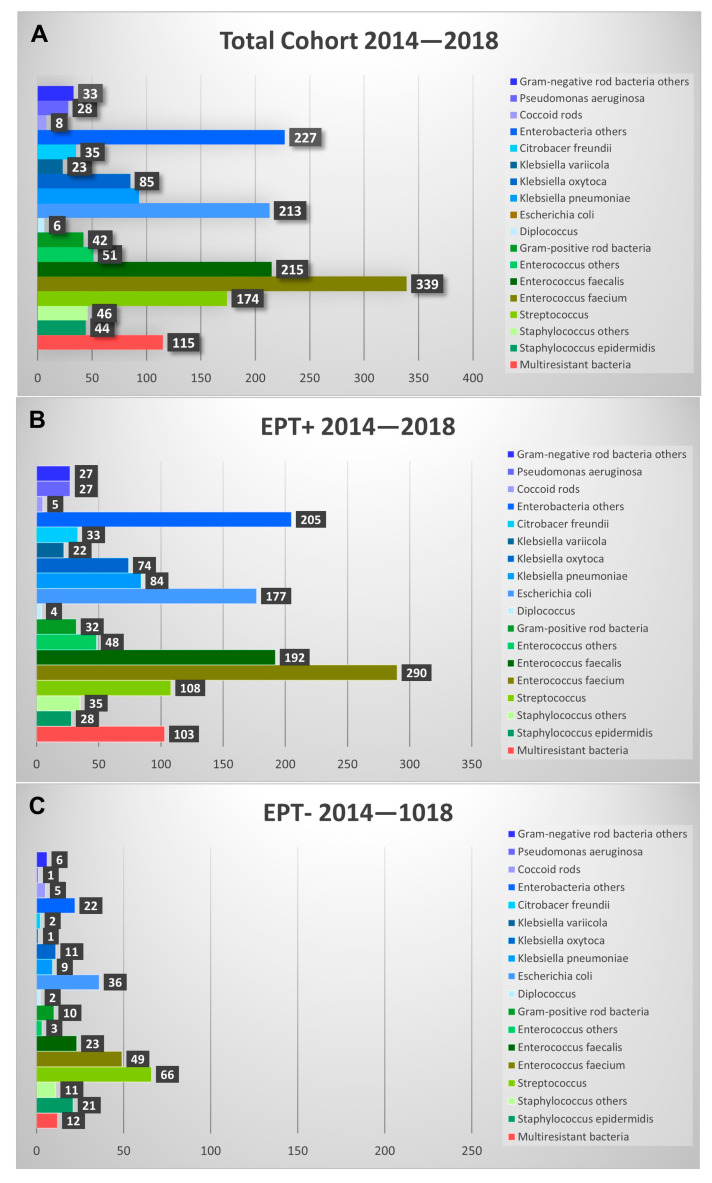
Shown are the different bacteria found in bile fluids during the study period. Absolute findings are shown in the total cohort (**A**) and presented for EPT+ (**B**) and EPT− (**C**) patients. Note that roughly three-times-more EPT+ bile aspirates were analyzed than EPT− aspirates.

**Figure 5 biomedicines-13-01255-f005:**
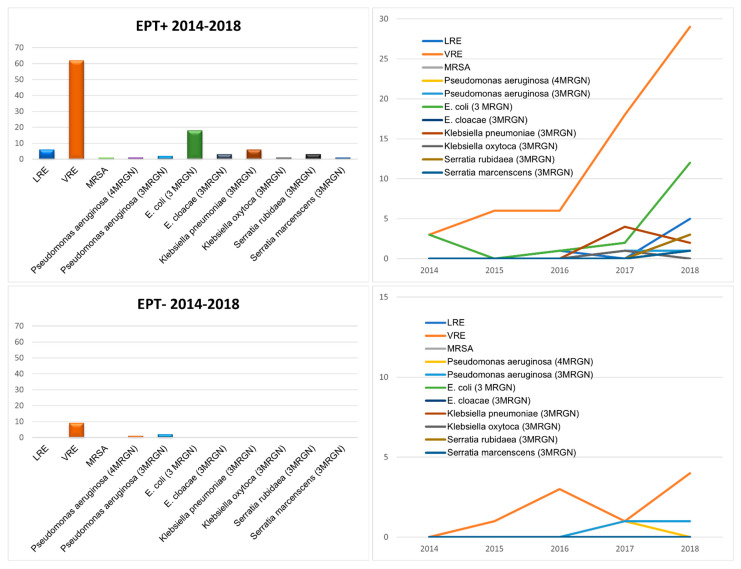
Multiresistant bacteria found in the bile fluids of the patients in this study, separated for EPT+ (first line) and EPT− (second line) patients. On the left side, absolute numbers during the study period are presented. On the right side, the increase in the different bacteria during the five years of the study period is shown.

**Figure 6 biomedicines-13-01255-f006:**
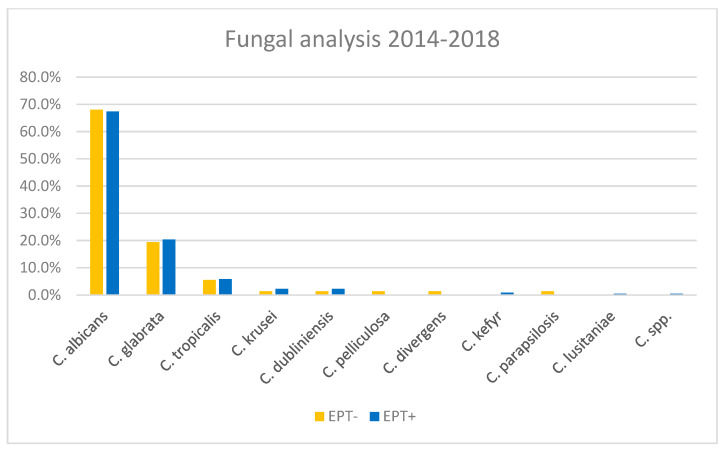
Percentages of the different *Candida* spp. found in bile cultures, separated for EPT+ and EPT− patients.

**Figure 7 biomedicines-13-01255-f007:**
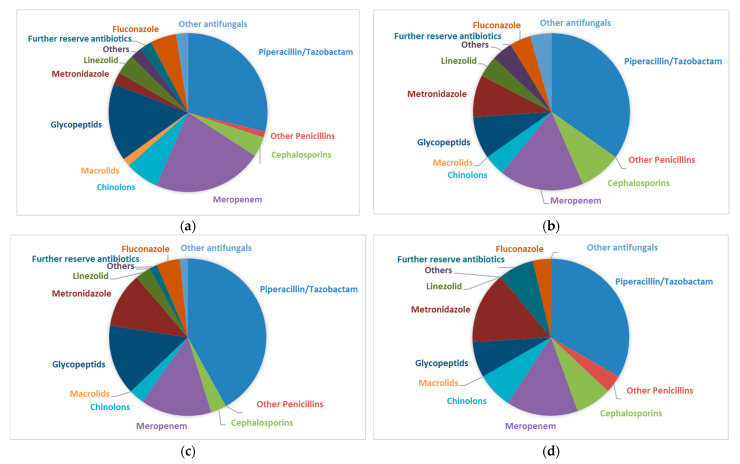
(**a**). Antibiotic regimens of patients with endoscopic sphincterotomy at a prior examination (EPT+) and positive cultures (MiBi pos) from bile aspirated during ERCP. Cephalosporinsincluded ceftriaxon (90%) and cefepim. Chinolons included ciprofloxacin (88%) and moxifloxacin. Glycopeptides included vancomycin (92%) and teicoplanin. Others included cotrimoxazol and clindamycin. Reserve antibiotics included tigecyclin, colistin, daptomycin, rifampicin, and fosfomycin. Other antifungalsincluded caspofungin (100%) and ambisome. (**b**). Antibiotic regimens of patients with endoscopic sphincterotomy at a prior examination (EPT+) and negative cultures (MiBi neg) from bile aspirated during ERCP. Cephalosprins included ceftriaxon. I Chinolons ncluded ciprofloxacin. Glycopeptides included vancomycin. Others included cotrimoxazol and clindamycin. Reserve antibiotics included tigecyclin, colistin, daptomycin, rifampicin, and fosfomycin. Other antifungals included caspofungin. (**c**). Antibiotic regimens of patients without endoscopic sphincterotomy at a prior examination (EPTi) and positive cultures (MiBi pos) from bile aspirated during ERCP. Cephalosporins included ceftriaxone. Chinolons included ciprofloxacin. Glycopeptides included vancomycin. Others included cotrimoxazol and clindamycin. Reserve antibiotics included tigecyclin, colistin, daptomycin, rifampicin, and fosfomycin. Other antifungals included ambisome. (**d**). Antibiotic regimens of patients without endoscopic sphincterotomy at a prior examination (EPT−) and negative cultures (MiBi neg) from bile aspirated during ERCP. Cephalosporins included ceftriaxon cefepime. Chinolons included ciprofloxacin. Glycopeptides included vancomycin. Others included cotrimoxazol and clindamycin. Reserve antibiotics included tigecyclin, colistin, daptomycin, rifampicin, and fosfomycin. Other antifungals included caspofungin and ambisome.

**Table 1 biomedicines-13-01255-t001:** Demographics and indications for ERC of the study population.

		Median [Years]	Range
**Age**		56.7	3 to 96 years
		**n =**	**[%]**
**Male**		1463	64.9%
**ERCs included**		2253	100%
**Bile aspirated**		1220	54.2%
**Bile analyzed**		1197	53.1%
**Positive bile culture for any microorgansim**		1029	45.7%
**Indication**			
	bile duct stones/biliary pancreatitis	443	19.7%
	cholangiocellular carcinoma	211	9.4%
	cholangitis/cholangiosepsis	99	4.4%
	pancreatic carcinoma	127	5.6%
	Hepatocellular carcinoma/liver metastasis	102	4.5%
	sclerosing cholangitis	336	14.9%
	anastomotic stricture after LTPL	275	12.2%
	post-surgical insufficiencies/strictures	181	8.0%
	others	479	21.3%

## Data Availability

The datasets generated and/or analyzed during the current study are available from the corresponding author on request.on request due to restrictions (e.g., privacy, legal or ethical reasons).

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
