# Peer review of "The Presence and Significance of Bacteria and Fungi in Bile Aspirated During ERC—A Retrospective Analysis"

_biomedicines, 2025, doi:10.3390/biomedicines13051255_

Round 1
Reviewer 1 Report
Comments and Suggestions for Authors
The presence and significance of bacteria and fungi in bile as- pirated during ERC - a retrospective analysis
Wiegand et al., in this manuscript present a well-structured study with relevant findings and potential clinical implications. Their study explores a significant biomedical issue. The methodology is generally sound and follows appropriate protocols. Figures and tables effectively illustrate the results. Despite these merits this manuscript has several flaws. I think some key weaknesses need to be addressed before submission to ensure clarity, rigor, and reproducibility.
For the starters, label needs to be refined.
Next, the abstract is too general and does not emphasize the novelty of the study. I’d clearly state the research gap the study addresses.
I think the key findings lack statistical significance values and confidence intervals. Please include specific data points such as effect sizes, p-values to highlight key results.
I’d also summarize the implications of the findings more explicitly.
The introduction does not adequately define the specific gap in the literature. I’d agree that the background information is descriptive but lacks critical analysis of prior work.
Please provide a clear research question and hypothesis statement at the end of the introduction.
It will be ideal if you could add references to recent studies to support the rationale.
Personally, I believe experimental design lacks clarity regarding:Sample selection criteria, Number of replicates per experiment, Statistical tests used and rationale for their selection
I think the controls are not well defined, making it difficult to assess validity.
Please clearly define control groups and their role in validating the results. I’d provide a detailed sample sizes and replicates to improve reproducibility. Finally, justify why specific statistical methods were chosen over others.
I feel like the results are not critically analyzed. Some statistical findings are ambiguous, lacking confidence intervals or effect sizes. Maybe expand the data interpretation beyond statistical significance.
I’d also provide biological explanations for observed trends. Please add confidence intervals to strengthen the credibility of results.
Some parts of the discussion repeat results instead of explaining their significance. The discussion does not compare findings with existing literature.
Study limitations are not addressed, which weakens transparency. So, maybe focus on why the results matter rather than restating them.
Then compare your findings with previous studies and highlight differences. Finally, add a Limitations section discussing potential biases or experimental challenges.
Some figures lack descriptive legends, making them difficult to interpret without reading the text. Graphs should indicate statistical significance such as asterisks for p-values. I’d suggest using Graph Pad prism for these figures where you can show replicates etc. Also, ensure figure legends are self-explanatory without requiring text reference. Label statistical significance clearly in graphs and tables.
The conclusion restates findings but does not emphasize future implications.
Finally, I’d highlight real-world applications and how this study advances the field.
And, suggest future research directions to strengthen impact.
Last but not the least, please pay special attention to grammar, comprehension and sentences. Breakdown complex sentences and avoid awkward phrasing.
Check the reference for volume and page numbers.
Italicise names, species of bacteria.
Best of luck and keep up the good work,
Cheers!
Comments on the Quality of English Language
The presence and significance of bacteria and fungi in bile as- pirated during ERC - a retrospective analysis
Wiegand et al., in this manuscript present a well-structured study with relevant findings and potential clinical implications. Their study explores a significant biomedical issue. The methodology is generally sound and follows appropriate protocols. Figures and tables effectively illustrate the results. Despite these merits this manuscript has several flaws. I think some key weaknesses need to be addressed before submission to ensure clarity, rigor, and reproducibility.
For the starters, label needs to be refined.
Next, the abstract is too general and does not emphasize the novelty of the study. I’d clearly state the research gap the study addresses.
I think the key findings lack statistical significance values and confidence intervals. Please include specific data points such as effect sizes, p-values to highlight key results.
I’d also summarize the implications of the findings more explicitly.
The introduction does not adequately define the specific gap in the literature. I’d agree that the background information is descriptive but lacks critical analysis of prior work.
Please provide a clear research question and hypothesis statement at the end of the introduction.
It will be ideal if you could add references to recent studies to support the rationale.
Personally, I believe experimental design lacks clarity regarding:Sample selection criteria, Number of replicates per experiment, Statistical tests used and rationale for their selection
I think the controls are not well defined, making it difficult to assess validity.
Please clearly define control groups and their role in validating the results. I’d provide a detailed sample sizes and replicates to improve reproducibility. Finally, justify why specific statistical methods were chosen over others.
I feel like the results are not critically analyzed. Some statistical findings are ambiguous, lacking confidence intervals or effect sizes. Maybe expand the data interpretation beyond statistical significance.
I’d also provide biological explanations for observed trends. Please add confidence intervals to strengthen the credibility of results.
Some parts of the discussion repeat results instead of explaining their significance. The discussion does not compare findings with existing literature.
Study limitations are not addressed, which weakens transparency. So, maybe focus on why the results matter rather than restating them.
Then compare your findings with previous studies and highlight differences. Finally, add a Limitations section discussing potential biases or experimental challenges.
Some figures lack descriptive legends, making them difficult to interpret without reading the text. Graphs should indicate statistical significance such as asterisks for p-values. I’d suggest using Graph Pad prism for these figures where you can show replicates etc. Also, ensure figure legends are self-explanatory without requiring text reference. Label statistical significance clearly in graphs and tables.
The conclusion restates findings but does not emphasize future implications.
Finally, I’d highlight real-world applications and how this study advances the field.
And, suggest future research directions to strengthen impact.
Last but not the least, please pay special attention to grammar, comprehension and sentences. Breakdown complex sentences and avoid awkward phrasing.
Check the reference for volume and page numbers.
Italicise names, species of bacteria.
Best of luck and keep up the good work,
Cheers!
Author Response
Commetns 1: The presence and significance of bacteria and fungi in bile aspirated during ERC - a retrospective analysis
Weigand et al., in this manuscript present a well-structured study with relevant findings and potential clinical implications. Their study explores a significant biomedical issue. The methodology is generally sound and follows appropriate protocols. Figures and tables effectively illustrate the results. Despite these merits this manuscript has several flaws. I think some key weaknesses need to be addressed before submission to ensure clarity, rigor, and reproducibility.
Response 1: We want to tank the reviewer for the many useful comments.
Comments 2: For the starters, label needs to be refined.
Response 2: Maybe the reviewer could make a suggestion what he would like to have refined?
Comments 3: Next, the abstract is too general and does not emphasize the novelty of the study. I’d clearly state the research gap the study addresses.
Response 3: We included a sentence with our clear aim to emphasize that we especially were interested in the influence of earlier sphincterotomy. Something that up to date never has been studied strongly.
Comments 4: I think the key findings lack statistical significance values and confidence intervals. Please include specific data points such as effect sizes, p-values to highlight key results.
Response 4: We understand the point. However, the data is mostly descriptive and as we mentioned in the figures with a clear trend statistical significance was not despite the increase of multiresistant bacteria. For that reasons we did not include p values.
Comments 5: I’d also summarize the implications of the findings more explicitly.
Response 5: We changed a sentence in the result section to emphasize the result oft he main goal. However, due tot he lack of significance it can not be said that earlier sphincterotomy is a risk for bacterial colonization.
Comments 6: The introduction does not adequately define the specific gap in the literature. I’d agree that the background information is descriptive but lacks critical analysis of prior work.
Response 6: We included more recent publications and findings in the introduction section.
Comments 7: Please provide a clear research question and hypothesis statement at the end of the introduction.
Response 7: Added
Comments 8: It will be ideal if you could add references to recent studies to support the rationale.
Response 8: We included further references in most sections of our manuscript.
Comments 9: Personally, I believe experimental design lacks clarity regarding: Sample selection criteria, Number of replicates per experiment, Statistical tests used and rationale for their selection
Response 9: We added some information to the M&M section to make sample collection more clear, Due to the retrospective analysis of our study there were no selection criteria. As we mentioned bile fluid collection was performed due tot he endoscopists choice but increased over time and become more or less routine in mostly every ERCP. If the sample was send to microbiological analysis was the choice oft he clinician. However samples were send for analysis in 98,1% as described in the M&M section.
Comments 10: I think the controls are not well defined, making it difficult to assess validity.
Response 10: We understand this critisim. Our study was a mainly descriptive retrospective analysis. Our main goal, as emphasized more in the manuscript thanks to the reviewers, was to compare patients without and with earlier sphincterotomy. So the patients without maybe considered „control group“.
Comments 11: Please clearly define control groups and their role in validating the results. I’d provide a detailed sample sizes and replicates to improve reproducibility. Finally, justify why specific statistical methods were chosen over others.
Response 11: As mentioned above, due to the retrospective analysis no „control groups“ could be defined. Comparing the above mentioned groups was the goal. The underlying diseases had no sigificant impact an the results as mentioned above. However, we did not intend to differentiate microbiological spectra related tot he underlying disease. Regarding statistical analysis we used the test as recommended by our institutes statistician. T test was used to compare mean values between to groups. To calculate the probability of identifying a bacterial species or group of bacteria between EPT+ and EPT- patients the Chi square test was used. Kolmogorov-Smirnov test was used to compare the distribution probability of two cohorts
Comments 12: I feel like the results are not critically analyzed. Some statistical findings are ambiguous, lacking confidence intervals or effect sizes. Maybe expand the data interpretation beyond statistical significance.
Response 12: As mentioned we were a bit disappointed that no statistical significance was found even with a highly powered study. We suspected higher frequency of positive bile cultures in the EPT+ over the EPT- group. Even with a tren towards more species found and a higher frequency in the EPT+ groups this failed to be statistical significant. Fort he above mentioned reason no confidence interval or p values were given. Due to the lack of significance we do not feel comfortable to expand our data interpretation.
Comments 13: I’d also provide biological explanations for observed trends. Please add confidence intervals to strengthen the credibility of results.
Response 13: See above. Our hypothesis was, that an altered sphincter leads to more frequent and easier ascendens of gut microbes into the bile duct. There was a trend, but unfortunately not significant.
Comments 14: Some parts of the discussion repeat results instead of explaining their significance. The discussion does not compare findings with existing literature.
Response 14: Thank you for pointing out that point. We included further literature and expanded our disucssion to further recent data
Comments 15: Study limitations are not addressed, which weakens transparency. So, maybe focus on why the results matter rather than restating them.
Response 15: We are sorry, but we do not understand this point. The penultimate paragraph of our discussion discusses only the limitations of our study which exist for sure. If the reviewer misses a limitation please point out and we are happy to include this as well.
Comments 16: Then compare your findings with previous studies and highlight differences. Finally, add a Limitations section discussing potential biases or experimental challenges.
Response 16: See above
Comments 17: Some figures lack descriptive legends, making them difficult to interpret without reading the text. Graphs should indicate statistical significance such as asterisks for p-values. I’d suggest using Graph Pad prism for these figures where you can show replicates etc. Also, ensure figure legends are self-explanatory without requiring text reference. Label statistical significance clearly in graphs and tables.
Response 17: See above
Comments 18: The conclusion restates findings but does not emphasize future implications.
Response 18: See above
Comments 19: Finally, I’d highlight real-world applications and how this study advances the field.
Response 19: We did so in the conclusions
Comments 20: And, suggest future research directions to strengthen impact.
Response 20: We added a future research suggestion at the end oft he manuscript
Comments 21: Last but not the least, please pay special attention to grammar, comprehension and sentences. Breakdown complex sentences and avoid awkward phrasing.
Response 21: Okay
Comments 22: Check the reference for volume and page numbers.
Response 22: Done
Comments 23: Italicise names, species of bacteria.
Response 23: Done
Comments 24: Best of luck and keep up the good work, Cheers!
Response 24: Thank you very much for the valuable comments
Reviewer 2 Report
Comments and Suggestions for Authors
The study was based on a retrospective analysis of 2,253 standard endoscopic retrograde cholangiography (ERC) procedures performed between 2014 and 2018. It focused on microbiological cultures of bile aspirates to identify the presence of bacteria and fungi in patients with biliary tract infections. The research examined the impact of factors such as sphincterotomy and prior antibiotic treatment on the frequency of positive cultures. The study’s findings emphasize the clinical relevance of microbiological analysis in guiding anti-infective therapy for biliary infections. However, major methodological and structural deficiencies critically undermine the manuscript’s validity and impact, necessitating significant revisions. If unresolved, rejection may be justified.
-
Introduction:
-
This section inadequately contextualizes the study. The introduction is incomplete and fails to address key topics relevant to the subject. A review of existing literature, the research gap, the study’s necessity, and its novelty are not clearly stated. These omissions weaken the scientific premise and hypothesis.
-
-
Materials and Methods:
-
The section is overly brief and lacks critical details. It should be expanded to provide comprehensive methodological information, including citations to support choices (e.g., ERC guidelines, culture protocols).
-
-
Line 80–83 (Clarity):
-
Original: “Patients were routinely anesthetised for the procedure with propofol and midazolam and monitored by pulse oxymeter and non-invasive blood pressure measurement during endoscopy and for at least 30 minutes after the end of the treatment.”
-
-
Line 121–122 (Clarity):
-
Original: “Because the main cause of cholangitis is ascending infection of the bile duct from the bowel, we investigated the influence of sphincterotomy on the number of positive microbiological bile samples.”
-
-
Figures:
-
The figures are presented in low resolution and inconsistent design, compromising data interpretation.
-
-
References:
-
Overreliance on outdated sources (References 3, 5–8, 11–12, 14, 19–20) reduces relevance to current clinical practices.
-
Conclusion:
The manuscript, in its current form, does not meet publication standards due to incomplete methodology, poor data presentation, and insufficient scientific context.
Author Response
The study was based on a retrospective analysis of 2,253 standard endoscopic retrograde cholangiography (ERC) procedures performed between 2014 and 2018. It focused on microbiological cultures of bile aspirates to identify the presence of bacteria and fungi in patients with biliary tract infections. The research examined the impact of factors such as sphincterotomy and prior antibiotic treatment on the frequency of positive cultures. The study’s findings emphasize the clinical relevance of microbiological analysis in guiding anti-infective therapy for biliary infections. However, major methodological and structural deficiencies critically undermine the manuscript’s validity and impact, necessitating significant revisions. If unresolved, rejection may be justified.
Comments 1: Introduction: This section inadequately contextualizes the study. The introduction is incomplete and fails to address key topics relevant to the subject. A review of existing literature, the research gap, the study’s necessity, and its novelty are not clearly stated. These omissions weaken the scientific premise and hypothesis.
Response 1: We added a paragraph in the introduction section addressing further literature and emphasizing the actual reseach need.
Comments 2: Materials and Methods: The section is overly brief and lacks critical details. It should be expanded to provide comprehensive methodological information, including citations to support choices (e.g., ERC guidelines, culture protocols).
Response 2: We added information regarding microbiological assessment and bile aspiration during ERCP.
Comments 3: Line 80–83 (Clarity): Original: “Patients were routinely anesthetised for the procedure with propofol and midazolam and monitored by pulse oxymeter and non-invasive blood pressure measurement during endoscopy and for at least 30 minutes after the end of the treatment.”
Response 3: We changed the sentence for clarity
Comments 4: Line 121–122 (Clarity): Original: “Because the main cause of cholangitis is ascending infection of the bile duct from the bowel, we investigated the influence of sphincterotomy on the number of positive microbiological bile samples.”
Response 4: We changed the sentence for clarity
Commemts 5: Figures: The figures are presented in low resolution and inconsistent design, compromising data interpretation.
Response 5: We were consistent in the colouring of the figures. We used the same color for earlier sphincterotomy or not in the associated figures. Furthermore we kept gram positive microbes in green, gram negative in blue and multiresistant microbes in red. However, we agree, that Figure 5 is hard to interprete due to only grey patterns. We provide the figure coloured.
Comments 6: References: Overreliance on outdated sources (References 3, 5–8, 11–12, 14, 19–20) reduces relevance to current clinical practices.
Response 6: We believe it is important to also provide earlier references to emphasize the reseach need and development of research in this field. However, we agree that there is some newer date and we included these references in our manuscript.
Comments 7: Conclusion: The manuscript, in its current form, does not meet publication standards due to incomplete methodology, poor data presentation, and insufficient scientific context.
Response 7: We are very sorry to receive this statement. We hope the changes made change this opinion.
Reviewer 3 Report
Comments and Suggestions for Authors
The original study represents a single-center report on retrospective bacteriological and fungal findings in bile specimens taken in standard manner by means of endoscopic retrograde cholangiography (ERC, 1220 bile samples) taken in patients with cholangitis, gallstones and other hepatobiliary and pancreatic disorders. The study covered a sufficient number of most clinical groups. Majority of the bile samples were found to be positive for bacteria, fungi or combined infection. Enterococci and Enterobacter were most common, whereas E. coli, Streptococci, Klebsiella and Staphylococci were also found, like as Candida spp. The results are interesting, due to description of typical bacterial landscape of bile detected by standard bacteriological cultures in routine clinical practice.
Remarks.
Line 16 (Abstract): The aim of study should be clearly formulated
Methods (Line 87). Standard bacteriological culture techniques should be described in brief (bacteriological media used, liquid and/or agar cultures, antibiotic sensitivity techniques), or appropriate guidelines can be referred.
Results
Line 147: A sentence is repeated …There was a trend to fewer different microorganisms in the bile of EPT- patients…. (Legend to Fig.2).
In Microbiological analysis (Line 150 and below): The percentages of culture-positive bile samples are shown to be relative stable with time. Also, it would be quite important to present relative frequencies of dominant bacteria in different disorders which are listed in Table 1. It would add a sufficient novelty to this extensive analysis.
Figures 2 and 3: Please skip the words: …Shown are the…
Line 129-130. It is not clear if bile sampling was in some cases repeatedly made (during and post-ERT), or the differences concern the whole studied cohort.
In this respect, comparative bacterial findings in repeatedly studied patients (if available) would be of sufficient interest (lines 173-174).
Discussion (Line 267) What means …. to rectify (may be, eliminate?) the cause….?
Line 275: … the potential effects of ERC on subsequent bacterial colonization are of sufficient interest. If such trend is detected, it may be stressed in Abstract and in Conclusion. In general, the text needs some copy editing to avoid problems with understanding of Results.
Comments on the Quality of English Language
Moderate copy editing is required
Author Response
The original study represents a single-center report on retrospective bacteriological and fungal findings in bile specimens taken in standard manner by means of endoscopic retrograde cholangiography (ERC, 1220 bile samples) taken in patients with cholangitis, gallstones and other hepatobiliary and pancreatic disorders. The study covered a sufficient number of most clinical groups. Majority of the bile samples were found to be positive for bacteria, fungi or combined infection. Enterococci and Enterobacter were most common, whereas E. coli, Streptococci, Klebsiella and Staphylococci were also found, like as Candida spp. The results are interesting, due to description of typical bacterial landscape of bile detected by standard bacteriological cultures in routine clinical practice.
Comments 1: Line 16 (Abstract): The aim of study should be clearly formulated
Response 1: We added a sentence in the abstract to clearly describe our aim
Comments 2: Methods (Line 87). Standard bacteriological culture techniques should be described in brief (bacteriological media used, liquid and/or agar cultures, antibiotic sensitivity techniques), or appropriate guidelines can be referred.
Response 2: Thank you for this remark. We added some information and especially references.
Comments 3: Results Line 147: A sentence is repeated …There was a trend to fewer different microorganisms in the bile of EPT- patients…. (Legend to Fig.2).
Response 3: Thanks for finding the duplication. We deleted one duplicate.
Comments 4: In Microbiological analysis (Line 150 and below): The percentages of culture-positive bile samples are shown to be relative stable with time. Also, it would be quite important to present relative frequencies of dominant bacteria in different disorders which are listed in Table 1. It would add a sufficient novelty to this extensive analysis.
Response 4: We agree that this is a very interesting question. However, we found no significant difference of detected bacteria related tot he underlying disease. For this reason we did not include the findings in our manuscript.
Comments 5: Figures 2 and 3: Please skip the words: …Shown are the…
Response 5: Deleted
Comments 6: Line 129-130. It is not clear if bile sampling was in some cases repeatedly made (during and post-ERT), or the differences concern the whole studied cohort.
Response 6: Thank you for this question. Bile aspiration was in every examination the primary action directly after intubation oft he main bile tract and was only done once in every single examination. We added this information in the M&M section and also changed the sentence within the text to make this more precise.
Comments 7: In this respect, comparative bacterial findings in repeatedly studied patients (if available) would be of sufficient interest (lines 173-174).
Response 7: Only a few patients really had repeatedly ERCs in this study. Within these patients bacterial spectrum remained unchanged, only multiresistant species increased over time. However, numbers are to small and fail any statistical relevance. Therefore, we did not include this data in our manuscript.
Comments 8: Discussion (Line 267) What means …. to rectify (may be, eliminate?) the cause….?
Response 8: Some causes can be eliminated, some have to be corrected, some can be removed or repaired. This always depends on the situation. This all is included …
Comments 9: Line 275: … the potential effects of ERC on subsequent bacterial colonization are of sufficient interest. If such trend is detected, it may be stressed in Abstract and in Conclusion. In general, the text needs some copy editing to avoid problems with understanding of Results.
Response 9: We rewrote some sentences for better understanding.